Zhang *et al. Genome Biology*　　(2020) 21:288

**RESEARCH**　　　　　　　　　　　　　　　　　　　　　　　　　**Open Access**

# Human A-to-I RNA editing SNP loci are enriched in GWAS signals for autoimmune diseases and under balancing selection

Hui Zhang[1,2†], Qiang Fu[1†], Xinrui Shi[1†], Ziqing Pan[1], Wenbing Yang[1], Zichao Huang[1], Tian Tang[3], Xionglei He[1] and Rui Zhang[1,4*]

* Correspondence: zhangrui3@mail.
sysu.edu.cn
[†]Hui Zhang, Qiang Fu and Xinrui
Shi contributed equally to this work.
[1]Key Laboratory of Gene
Engineering of the Ministry of
Education, State Key Laboratory of
Biocontrol, School of Life Sciences,
Sun Yat-Sen University, Guangzhou,
People's Republic of China
[4]RNA Biomedical Institute, Sun
Yat-Sen Memorial Hospital, Sun
Yat-Sen University, Guangzhou,
People's Republic of China
Full list of author information is
available at the end of the article

## Abstract

**Background:** Adenosine-to-inosine (A-to-I) RNA editing plays important roles in diversifying the transcriptome and preventing MDA5 sensing of endogenous dsRNA as nonself. To date, few studies have investigated the population genomic signatures of A-to-I editing due to the lack of editing sites overlapping with SNPs.

**Results:** In this study, we applied a pipeline to robustly identify SNP editing sites from population transcriptomic data and combined functional genomics, GWAS, and population genomics approaches to study the function and evolution of A-to-I editing. We find that the G allele, which is equivalent to edited I, is overrepresented in editing SNPs. Functionally, A/G editing SNPs are highly enriched in GWAS signals of autoimmune and immune-related diseases. Evolutionarily, derived allele frequency distributions of A/G editing SNPs for both A and G alleles as the ancestral alleles are skewed toward intermediate frequency alleles relative to neutral SNPs, a hallmark of balancing selection, suggesting that both A and G alleles are functionally important. The signal of balancing selection is confirmed by a number of additional population genomic analyses.

**Conclusions:** We uncovered a hidden layer of A-to-I RNA editing SNP loci as a common target of balancing selection, and we propose that the maintenance of such editing SNP variations may be at least partially due to constraints on the resolution of the balance between immune activity and self-tolerance.

**Keywords:** A-to-I RNA editing, Autoimmune and immune-related diseases, Transcriptome, Balancing selection

## Introduction

RNA editing is a process through which the sequence of an RNA is post-transcriptionally altered from that encoded in the DNA [1, 2]. A-to-I RNA editing is the most common type of RNA editing in metazoans [3]. It is mediated by Adenosine Deaminases Acting on RNA (ADARs), which bind dsRNA regions of protein-coding and non-coding RNAs and deaminate adenosine to inosine [2, 4, 5]. Inosine pairs

preferentially with cytidine, as opposed to uridine; therefore, editing alters the sequence and base-pairing properties of both protein-coding and non-coding RNAs. Editing of protein-coding genes may lead to nonsynonymous substitutions, and editing in non-coding RNAs or non-coding parts of protein-coding genes may regulate the splicing and stability of mRNA via multiple mechanisms [6, 7]. Furthermore, it has recently been shown that ADAR1-mediated editing of endogenous dsRNAs, particularly those in the non-coding regions, could disrupt the structures of dsRNA that were potentially bound by MDA5 and block MDA5-mediated immune response; therefore, RNA editing is required to prevent activation of the cytosolic innate immune system. This is most probably the essential function of ADAR1 editing [8–10].

Recent genome-wide searches of editing sites revealed thousands to millions of A-to-I editing events in various species [3]. However, identification of RNA editing sites is still hampered by the difficulty to distinguish true editing sites from A/G genomic variants, especially for samples without matched genomic and transcriptomic data. To minimize false positives arising from genomic variants, the editing identification pipelines that have been recently developed by us and other groups generally discard sites overlapping with known SNPs (e.g., [11–13]). Consequently, previous evolution studies focused on non-SNP editing sites, particularly those that lead to nonsynonymous changes [14–20]. To date, little is known about the population genomic signature of RNA editing. Before the next-generation sequencing era, a pioneer study has identified A-to-I RNA editing sites in the SNP database [21]. In many cases, SNPs overlapped with editing sites are annotated using expressed sequence tags, and thus are RNA editing sites instead of SNPs. However, it is possible that some of these SNPs are real SNPs that can be edited. And such editing SNPs are of importance for functional and evolutionary studies of RNA editing.

Here, we modified our previous approach to achieve robust identification of both non-SNP editing sites and SNP editing sites (editing sites overlapped with SNPs) for samples with matched genomic and transcriptomic data. We applied this approach to samples from Genotype-Tissue Expression (GTEx) and Geuvadis projects and identified SNP editing sites to study the function and evolution of RNA editing in humans.

## Results

### Identification of SNP editing sites

To generate the list of SNP editing sites, we modified our previous pipeline [11, 22] to retain RNA variants overlapping with known SNPs for editing site calling (Additional file 1: Fig. S1), and applied the modified pipeline to both GTEx and Geuvadis datasets. The GTEx v7 dataset contains a total of 11,688 samples, and we selected the transcriptomes of 3315 human samples (representing 27 tissue types, Additional file 2: Table S1) for analysis. The Geuvadis dataset contains transcriptome data from 464 immortalized B cell line samples. To identify editing sites with high confidence, we only selected samples in which the proportion of A-to-G/T-to-C variants to total variants were at least 80% for editing site call [11, 22]. Editing sites that were overlapped or non-overlapped with SNPs were called separately (Fig. 1a and Additional file 1: Fig. S2a). For the GTEx dataset, we identified 6407 SNP editing sites and 259,462 non-SNP editing sites. For the Geuvadis dataset, we identified 1651 SNP editing sites and 34,419

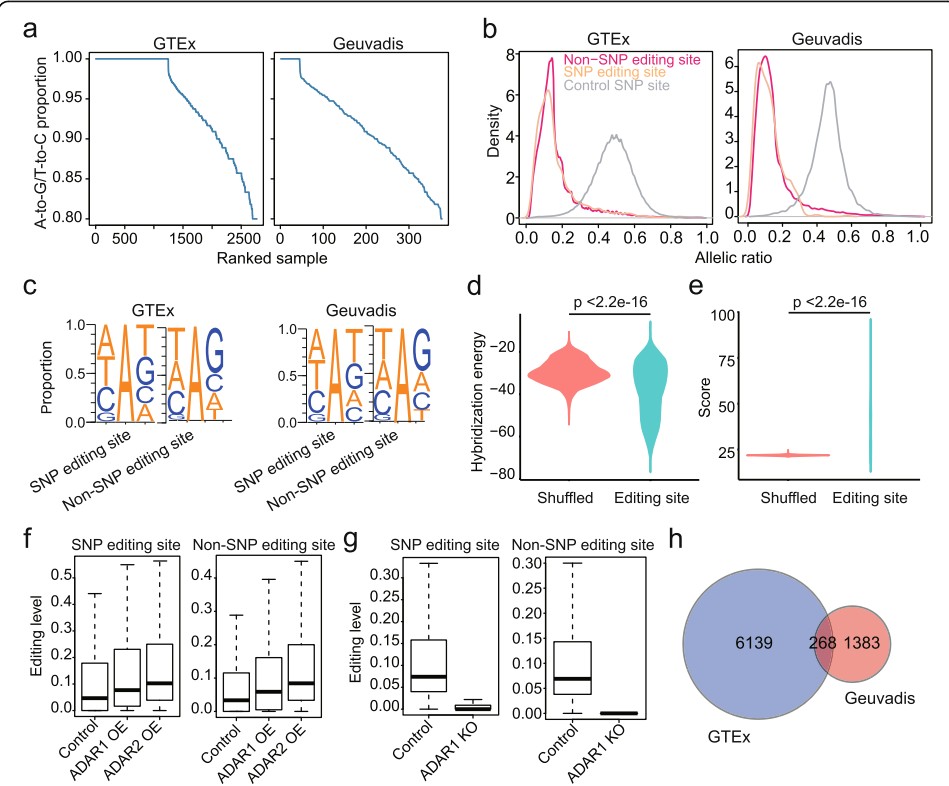

**Fig. 1** The identification and verification of non-SNP and SNP editing sites. **a** The proportion of variants that are either A-to-G or T-to-C mismatches for SNP RNA variants in each sample. The GTEx and Geuvadis datasets were plotted, separately. All samples used for editing identification were shown. **b** Allelic ratio distributions for non-SNP editing sites, SNP editing sites, and control SNPs. SNP and non-SNP editing sites called in all individuals were plotted. Control SNPs are all known SNPs. For each control SNP, the allelic ratios for individuals with heterozygous genotype were calculated. Only samples with sites that are covered by at least 20 reads were used. **c** The nucleotides neighboring both the non-SNP and SNP editing sites show a pattern consistent with known ADAR preference. The motif is characterized by the underrepresentation of G upstream to the editing site. **d** Comparison of the hybridization energies between SNP editing regions (SNP editing sites and flanking ± 15 nt) and their predicted complementary sequences with those between shuffled editing regions and their predicted complementary sequences (see the "Methods" section). For each SNP editing site, we shuffled the editing region and predicted its complementary sequence. We repeated this 10,000 times and the mean value was calculated. The *p* value was calculated with the pairwise Wilcoxon rank sum test. **e** Comparison of the BLAST scores between SNP editing regions (SNP editing sites and flanking ± 25 nt) and shuffled editing regions. BLAST score represents the overall quality of an alignment (aligning the editing region to the genomic sequence ± 2000 nt of the SNP editing site, see the "Methods" section). The *p* value was calculated with the pairwise Wilcoxon rank sum test. **f** Boxplots showing the editing level changes of SNP editing sites and non-SNP editing sites upon ADAR1 or ADAR2 overexpression in HEK293 cells. **g** Boxplots showing the editing level changes of SNP editing sites and non-SNP editing sites between wild-type and ADAR1 knockout HEK293 cells. **h** The overlaps of SNP editing sites between GTEx and Geuvadis datasets

non-SNP editing sites (Additional file 3: Table S2). The proportions of SNP editing sites in the GTEx and Geuvadis datasets are 0.024 and 0.046, respectively. The number of identified SNP editing sites increased with increasing sample size and remained unsaturated (Additional file 1: Fig. S2b), suggesting the presence of a hidden layer of editing sites that were previously ignored in the human genome. As a control, we applied the same pipeline to call C-to-T/G-to-A editing SNPs in the GTEx and Geuvadis datasets. We identified 11 and 0 sites (including 3 and 0 non-Alu sites in the CDS regions),

which were much less than the 6407 and 1651 A-to-G sites we identified (including 460 and 68 non-Alu sites in the CDS regions).

To validate whether the identified editing sites are bona fide A-to-I editing events, we examined their editing levels in comparison with RNA allelic ratios of known human SNPs. We found that, for both non-SNP and SNP editing sites, the distributions of their editing levels differed from allelic ratios of known heterozygous SNPs, which were centered at 0.5 (Fig. 1b). We also examined RNA editing triplet motifs for non-SNP and SNP editing sites. We found that both were associated with the underrepresentation of guanosines immediately 5′ of the edited adenosine (Fig. 1c), consistent with the known ADAR preference [23, 24]. When examining non-Alu CDS and other SNP editing sites separately, we found that non-Alu CDS sites had a slightly weaker ADAR motif than other sites (Additional file 1: Fig. S3). These analyses support that both non-SNP and SNP RNA variants we called were enriched in authentic editing events. In addition, more caution is needed when investigating individual non-Alu CDS sites since they may have a higher false-discovery rate than sites in other genic regions.

To ask whether SNP editing sites are enriched in regions with dsRNA structures required for ADAR activity, we performed two analyses. First, we predicted the editing complementary sequence (ECS) of each editing region (SNP editing site and flanking ± 15 nt) and the shuffled editing region, as previously described [25]. Next, we compared the hybridization energies between SNP editing regions and their predicted ECSs with those between shuffled editing regions and their predicted ECSs. We found significantly lower hybridization energies of editing regions than the shuffled regions (Fig. 1d). Moreover, about 44% of the SNP editing sites had a statistically significant ECS (see the "Methods" section). Second, we detected the potential dsRNA structures containing SNP editing sites using bl2seq, as previously described [26]. We found that the editing regions formed dsRNA structures with significantly higher alignment scores as compared to the shuffled regions (Fig. 1e). The same analyses were performed for non-Alu SNP editing sites, and the conclusions still held (Additional file 1: Fig. S4a-b).

Finally, to experimentally verify that SNP editing sites are real A-to-I editing events, we examined their editing level changes upon ADAR1 or ADAR2 overexpression in HEK293 cells [27]. We found that both non-SNP and SNP editing sites had increased levels upon overexpression of individual ADARs (Fig. 1f). We also examined ADAR1 knockout HEK293 cells [28] and ADAR1 or ADAR2 knockdown B cells [29]. As expected, both non-SNP and SNP editing sites had decreased editing levels in the knockout or knockdown cells (Fig. 1g and Additional file 1: Fig. S4c-d).

Having proved the validity of SNP editing sites, as many SNP editing sites were not overlapped between the two datasets (Fig. 1h), possibly due to the different origins of tissue types, we merged the two lists and obtained a total of 7790 SNP editing sites, including 278 recoding sites, for analysis.

### Characterizing SNP editing sites

A comparison between non-SNP and SNP editing sites revealed the difference in their genic locations and repetitive sequence features. SNP editing sites tended to be in CDS regions and 3′UTR regions, while non-SNP editing sites tended to be in the intergenic regions and intronic regions (Fig. 2a). In addition, compared with non-SNP editing

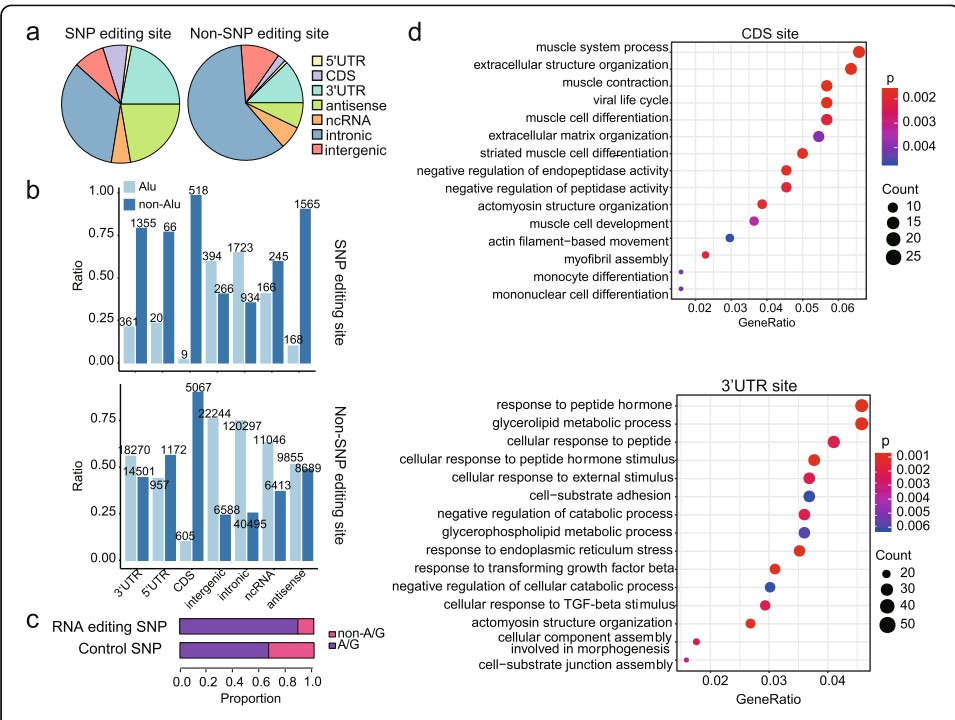

**Fig. 2** The characterization of non-SNP and SNP editing sites. **a** The genic locations of non-SNP and SNP editing sites. CDS, coding DNA sequence; ncRNA, non-coding RNA; intronic, intron of protein-coding genes and non-coding RNA; antisense, antisense RNAs. **b** The proportion of SNP and non-SNP editing sites in Alu and non-Alu regions of different genic locations. Numbers of editing sites are listed above the bars. **c** Comparison of the SNP types between editing SNPs and control SNPs. All SNPs in the 1000 Genomes Project that are with A or T as the reference allele were selected as control SNPs. **d** GO term enrichment for genes that contain SNP editing sites in CDS regions or 3′UTR regions, respectively

sites, SNP editing sites tended to be in non-Alu regions in all genic locations studied (Fig. 2b). The densities of editing SNPs varied among different functional classes, and such difference is not due to the difference of background SNP densities in different functional classes (Additional file 1: Fig. S5). These findings suggest that SNP editing sites may be functionally important. In line with this, when examining the types of SNPs in SNP editing sites, we found that editing SNPs were biased toward A/G or T/C (for genes in the forward or reverse strand) genotypes compared with the control SNPs (Fig. 2c). This result suggests that the G allele, which is functionally equivalent to edited I, was selected to be maintained in SNP editing sites.

To ask the possible functional significance of SNP editing sites, we performed Gene Ontology (GO) term analysis. As editing sites in CDS regions and 3′UTR regions may lead to different functional consequences, these two sets of sites were analyzed, separately. We found that genes containing CDS SNP editing sites were highly enriched in muscle-related functions (Fig. 2d). This result is in line with the finding that ADAR2, which is the primary editor of nonrepetitive coding sites [30], had the highest expression level in the artery (Additional file 1: Fig. S6a). For example, a nonsynonymous SNP editing site was identified in gene CASQ2 (muscle contraction GO term). CASQ2 is highly expressed in the heart and artery (Additional file 1: Fig. S6b) and involved in the storage and transport of positively charged calcium atoms [31]. CASQ2 plays an integral role in cardiac regulation, and its mutations have been associated with cardiac

arrhythmia and sudden death [32]. In contrast, genes containing 3′UTR SNP editing sites, which are likely ADAR1 targets, were highly enriched in functional categories such as cellular response to ER stress or external stimulus (Fig. 2d).

Taken together, these results suggest a hidden layer of A-to-I RNA editing events that are likely functionally important.

### The link between SNP editing sites and autoimmune and immune-related functions

Genome-wide association studies (GWAS) have led to the identification of thousands of SNPs linked to disease susceptibility in complex human diseases [33]. Based on our finding that in most cases the SNP type of editing SNPs is equivalent to the editing type (A/G SNP vs A-to-I editing), to characterize the potential phenotypic effect of SNP editing sites, we utilized GWAS data and examined the enrichment of editing SNPs in human disease GWAS. Interestingly, we found that editing SNPs are highly enriched in GWAS signals for autoimmune (for example, inflammatory bowel disease (IBD) and Crohn's disease) and immune-related diseases (e.g., coronary artery disease [34]) (Fig. 3a), suggesting that RNA editing may play an important role in autoimmune and immune-related functions. This agrees with the fact that (1) the major biological function of RNA editing is to suppress dsRNA-mediated autoimmunity, and (2) ADAR1 loss-of-function and MDA5 gain-of-function mutations are identified in autoimmune diseases [8–10].

We next compared the editing levels between non-SNP editing sites, SNP editing sites, and GWAS SNP editing sites, which may be informative for assessing their biological significance. We found that SNP and GWAS SNP editing sites in CDS regions had higher editing levels than non-SNP editing sites, while SNP editing sites in intronic or intergenic regions had similar editing levels as compared with non-SNP editing sites (Fig. 3b). Thus, it seems that SNP editing sites in functionally important regions tended to have higher editing levels.

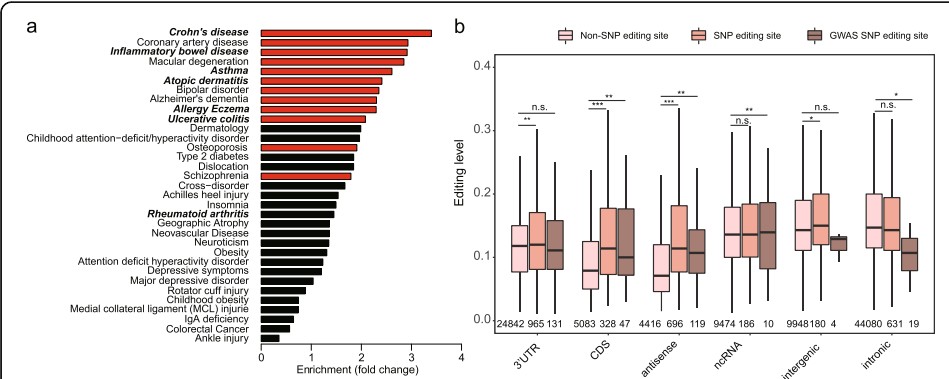

**Fig. 3** Editing SNPs are enriched in GWAS loci associated with immune-related diseases. **a** Enrichment analysis for 33 human diseases. Autoimmune diseases are shown in bold italic. *p* values were calculated with Fisher's exact test, and the significant ones are labeled with red. **b** Comparison of editing levels between SNP editing sites, non-SNP editing sites, and GWAS SNP editing sites. 5′UTR sites are not shown because only 3 GWAS SNP sites were found. For this analysis, we used the representative editing level of each editing site, which is the maximum editing level across all GTEx tissue types we profiled. The editing level of a tissue is the mean editing level of all samples in a given tissue. *p* values were calculated with the Mann-Whitney *U* test

### Derived allele frequency (DAF) analysis reveals RNA editing as the target of balancing selection

Balancing selection is the main force shaping the evolution of immunity genes [35–38]. The finding that RNA editing is enriched in loci related to autoimmune and immune-related functions, along with the putative functional difference between the A and I/G alleles, suggested that SNP editing loci may play a role in balancing immune system. If this is the case, evolutionarily, we predicted that a signal of balancing selection in SNP editing loci, particularly for the A/G SNP type that is equivalent to the A-to-I editing, would be found.

To verify our prediction, we applied population genomic approaches to study the evolution of RNA editing in humans. First, we examined DAF distributions of editing SNPs. A/G and non-A/G editing SNPs were analyzed separately, as they may be subject to different evolutionary constraints. To prevent the ascertainment bias between functional classes, CDS region, UTR of protein-coding genes, and ncRNAs were respectively analyzed. Different DAF distributions were directly compared using the Kolmogorov-Smirnov test, as previously described [39]. Intriguingly, we found that the DAF distribution of A/G editing SNPs was significantly skewed toward intermediate frequency alleles in all functional classes relative to intergenic regions (Fig. 4a, Additional file 1: Fig. S7, *p* values in Additional file 4: Table S3). The excess of intermediate frequency alleles, which is a typical scenario of balancing selection, implies that A/G editing SNPs were under balancing selection. In contrast, a shift in a DAF distribution toward low frequency alleles was observed for non-A/G editing SNPs, which is indicative of negative selection (Fig. 4b, Additional file 1: Fig. S7, *p* values in Additional file 4: Table S3). To ask whether there are different evolutionary patterns for editing SNPs with editable (A allele) or un-editable allele (non-A allele) as the ancestral allele, we examined their DAF distributions separately. In A/G editing SNPs, we found that both were under balancing selection (Fig. 4c, d, Additional file 1: Fig. S7, *p* values in Additional file 4: Table S3). In non-A/G editing SNPs, we found that, for SNPs with editable allele as the ancestral allele, the SNPs were under negative selection (Fig. 4e, Additional file 1: Fig. S7, *p* values in Additional file 4: Table S3). SNPs with un-editable allele as the ancestral allele were not analyzed because only a dozen sites were identified. Finally, we performed the DAF analysis for the upstream and downstream SNPs of non-SNP editing sites. Known editing sites from RADAR2 database [40] and non-SNP editing sites identified in this study were merged for analysis. As expected, no signatures of balancing selection were observed (Fig. 4f, Additional file 1: Fig. S7, *p* values in Additional file 4: Table S3), suggesting that the selection was specific to SNP editing sites. Taken together, these data suggest that the A/G editing SNPs were under balancing selection.

### Additional evidence to support the balancing selection on A-to-I RNA editing

In addition to the DAF analysis, we performed additional analyses to confirm A/G editing SNPs as targets of the balancing selection. We found that the sequence variation surrounding the SNP editing sites fulfills two predictions from population genetic theory for genomic regions either directly experiencing long-term balancing selection or genetically linked to them. The first prediction is that we expect editing SNP regions to exhibit an excess of intermediate frequency alleles compared with the expectation

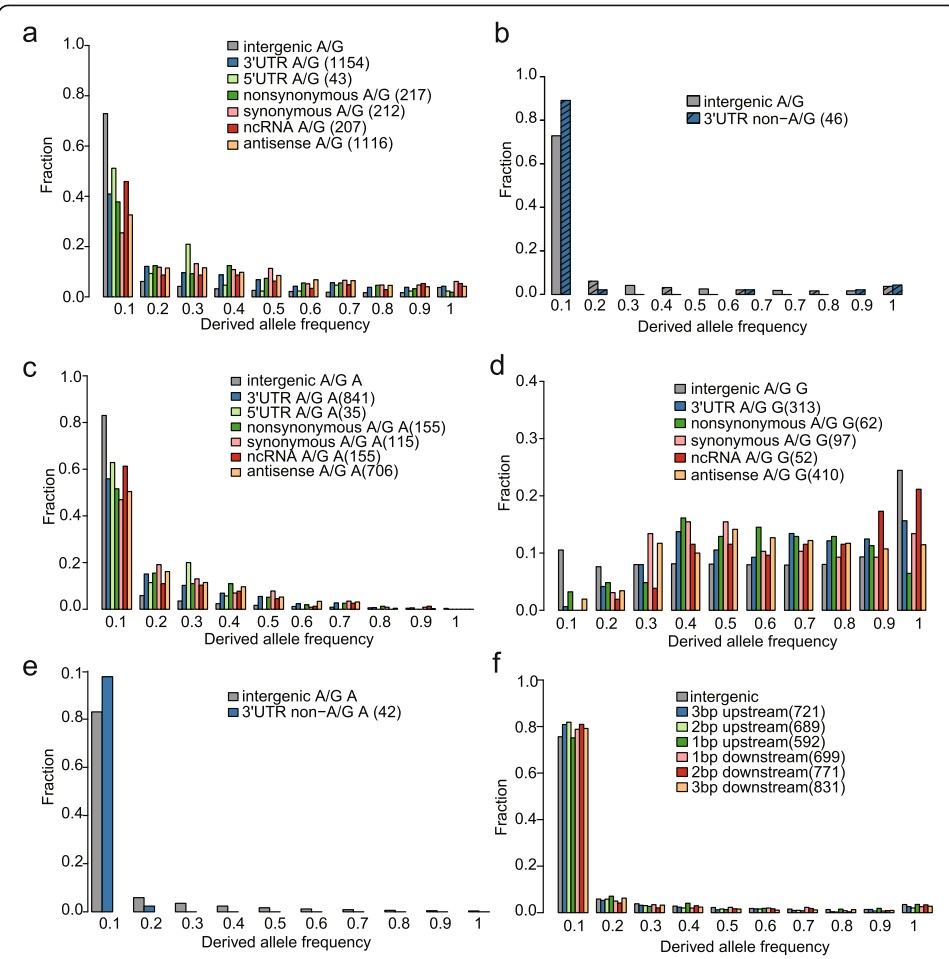

**Fig. 4** DAF analysis reveals RNA editing SNPs as the target of balancing selection. **a**, **b** DAF distributions of A/G (**a**) or non-A/G (**b**) editing SNPs. For non-A/G editing SNPs, only SNPs located in 3′UTR had enough number for the analysis. DAF was calculated using the genotype data from the 1000 Genomes Project. *p* values were calculated with the Kolmogorov-Smirnov test by comparing the DAF distribution of editing SNPs in a defined genic location with the distribution of SNPs in intergenic regions (Additional file 4: Table S3). The numbers of RNA editing SNPs in each genic location are shown in parentheses. Intergenic-A/G or Intergenic-non-A/G: all A/G or non-A/G SNPs located in the intergenic regions as control. **c**, **d** DAF distributions of A/G editing SNPs with A (**d**) or G (**e**) allele as the ancestral allele. In **d**, 5′UTR sites were not analyzed due to the limited number. **e** DAF distributions of non-A/G editing SNPs. Only SNPs that are with A allele as the ancestral allele and located in 3′UTR had enough number for this analysis. **f** DAF distributions of the SNPs located in the upstream and downstream of 3′UTR non-SNP editing sites. Intergenic, all SNPs located in intergenic regions

under selective neutrality. This is indicated by positively skewed values of Tajima's *D* (a summary of the mutation site-frequency spectrum [41]) of editing SNP regions, in comparison with the control regions (Fig. 5a). The second prediction is that neutral variants linked to the site under balancing selection are expected to be maintained in a population and generate excess diversity around the target of selection. The excess of high pi (nucleotide diversity within species) values of editing SNP regions, compared with the control SNP regions, is consistent with this prediction (Fig. 5b). Moreover, a sliding window analysis revealed that, compared with both the flanking regions and the control A/G SNP regions, editing SNP regions had higher pi values (Fig. 5c and Additional file 1: Fig. S8).

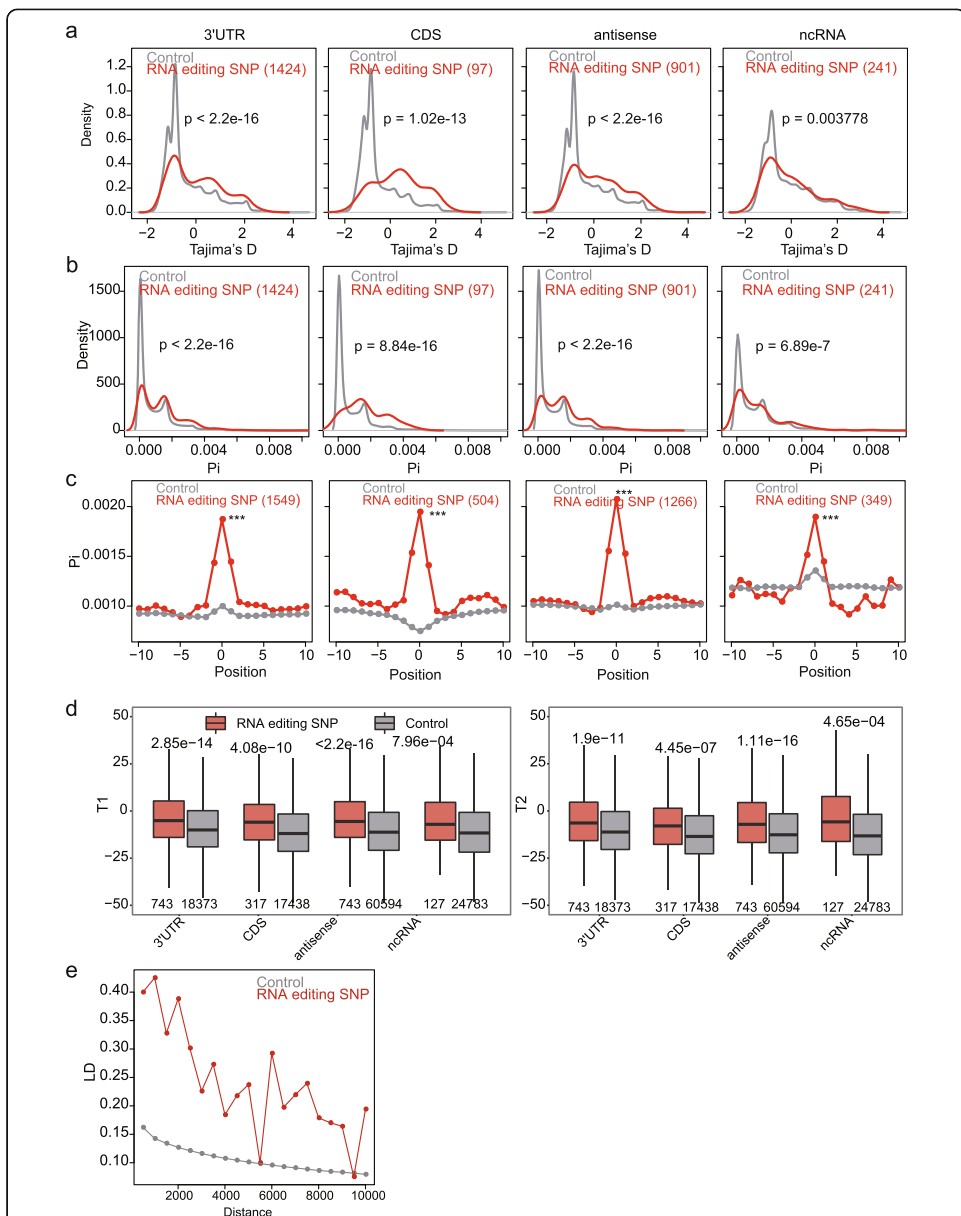

**Fig. 5** Additional analyses to support the balancing selection on RNA editing SNPs. **a**, **b** The distribution of Tajima's *D* values (**a**) and pi (**b**) for A/G editing SNPs and control SNPs. Control SNPs are A/G SNPs located at the same genic locations as the A/G editing SNPs compared. The control SNPs in **c–e** are the same as in **a**. *p* values were calculated using the Mann-Whitney *U* test. 5′UTR editing SNPs were not analyzed due to the limited number. **c** Sliding window analysis of pi for A/G editing SNPs and flanking regions. A 200-bp sliding window (step size = 100 bp) was used, and the average pi in each window was shown. Moreover, a comparison between the A/G editing SNP containing window and the control A/G SNP containing window was performed. *p* values were calculated using the Mann-Whitney *U* test. ***$p$ < 0.001. **d** Comparison of T1 and T2 scores between A/G editing SNPs and control SNPs. *p* values were calculated using the Mann-Whitney *U* test. **e** LD ($r^2$) between pairs of A/G editing SNPs (red) and pairs of control SNPs (gray). The flanking SNPs were assigned into 20 bins (from 500 to 10,000 bp) based on their distance to editing SNPs or control SNPs. The average LD in each bin was calculated and plotted

Recently, DeGiorgio et al. proposed two model-based summary statistics (T1 and T2, which generate a composite likelihood of a site being under balancing selection) to detect balancing selection [42]. In this method, sites having higher scores are more likely

to be under balancing selection. When comparing these summary statistics between editing SNPs and control SNPs, we found that, as expected, the editing SNPs had higher scores than the control SNPs for both T1 and T2 statistics (Fig. 5d and Additional file 5: Table S4).

In addition to the elevated polymorphism in editing SNP regions, we also found evidence for increased LD (another hallmark of balancing selection). We compared local LD (< 10,000 bp, measured as $r^2$) between pairs of RNA editing SNPs and pairs of control SNPs. Consistent with balancing selection, we found that pairs of RNA editing SNPs had higher LD than pairs of control SNP sites (Fig. 5e).

To ask whether both Alu and non-Alu editing SNPs are under balancing selection, we repeated the analyses above for the two groups of SNPs, separately. We found that non-Alu editing SNPs were more biased toward A/G genotypes than Alu editing SNPs (Additional file 1: Fig. S9a). DAF distribution of non-Alu editing SNPs was significantly skewed toward intermediate frequency alleles relative to Alu editing SNPs (Additional file 1: Fig. S9b). Consistent with this finding, the comparison of Tajima's *D* and pi values, as well as the T1 and T2 scores, between non-Alu editing SNPs and Alu editing SNPs, all suggested that non-Alu editing SNPs were subject to a stronger balancing selection signal than Alu editing SNPs (Additional file 1: Fig. S9c-e).

### DAF analysis supports RNA editing as the target of balancing selection in flies

Finally, to ask whether editing SNPs are under balancing selection in other species, we examined *Drosophila melanogaster* data. The genotype data from the Drosophila Genetics Reference Panel Project (DGRP) [43], which consists 205 sequenced inbred lines derived from Raleigh (NC), USA, were used to perform SNP allele type and DAF analysis. We examined all known RNA editing sites from RADAR2 database [40], which were called from three other *D. melanogaster* strains (w1118, Canton-S, and OregonR). A total of 743 sites were found to be overlapped with SNPs in DGRP. Similar to human, fly editing SNPs were biased toward A/G or T/C genotypes as compared with the control SNPs (Additional file 1: Fig. S10a). Moreover, the DAF distribution of A/G editing SNPs was significantly skewed toward intermediate frequency alleles in all functional classes relative to intergenic regions (Additional file 1: Fig. S10b-c, *p* values in Additional file 4: Table S3). In contrast, a shift in a DAF distribution toward low frequency alleles was observed for non-A/G editing SNPs, which is indicative of negative selection (Additional file 1: Fig. S10d-e, *p* values in Additional file 4: Table S3). These results support RNA editing as the target of balancing selection in flies.

### Discussion

Balancing selection is a mode of adaptation that leads to the maintenance of variation in a species and potentially an important biological force for maintaining advantageous genetic diversity in populations [44–47]. Immunity genes are known as the targets of balancing selection due to the need of immune system genetic plasticity in response to various stimuli, such as different pathogens [35–38]. In this study, we revealed that, unexpectedly, a previously unidentified type of variants (i.e., SNPs in A-to-I RNA editing sites) is the common target of balancing selection in humans. Based on the known functions of ADAR1 and the observed enrichment of editing SNPs in GWAS signals

for autoimmune and immune-related diseases, it is likely that editing SNP variations are maintained at least partially because of constraints on the resolution of the balance between immune activity and self-tolerance.

Several studies have found that editable A's are more likely to be replaced with G's during evolution [15, 48–50]. Some researchers think that this phenomenon suggests that A-to-I RNA editing serves as a safeguard and is beneficial because it reverses harmful G-to-A mutation in RNA transcripts; others argue that this observation suggests that A-to-I RNA editing is non-adaptive because G's are more acceptable at the editable A sites than un-editable A sites. As we found that within the population, A/G alleles at the editing loci were under balancing selection and beneficial, our findings suggest that the previous observation between species was made because the two alleles can be favorable in different conditions, which leads to the fixation of the G allele in some species and the maintenance of the A allele in other species.

## Conclusion

In summary, we uncover a hidden layer of human A-to-I editing SNP loci that are of functional importance, enriched in GWAS signals for autoimmune diseases, and subject to balancing selection. Various types of RNA editing, including A-to-I editing, alter sequence relative to the genome at the RNA level, thus providing a rich resource of RNA variants that potentially produce functionally altered genes. For some of the RNA variants that are beneficial under certain conditions, once the same type of mutation occurs at the DNA level, it may be selectively maintained and become the target of balancing selection. Therefore, we hypothesized that RNA editing, as exemplified in this study with A-to-I editing, may be an unrecognized type of the common target of balancing selection in various species.

## Methods

### RNA-seq data collection

Geuvadis RNA-seq data were downloaded from https://www.ebi.ac.uk/Tools/geuvadis-das/. GTEx project data were downloaded from NCBI, and a list of GTEx data sample IDs is shown in Additional file 2: Table S1.

### Mapping of RNA-seq reads

We adopted a pipeline that can accurately map RNA-seq reads to the genome [11]. In brief, we used BWA [51] to align RNA-seq reads to a combination of the reference genome and exonic sequences surrounding known splicing junctions from available gene models. We chose the length of the splicing junction regions to be 1 bp shorter than the RNA-seq reads to prevent redundant hits. We obtained gene models from UCSC genome browser: a combination of Gencode, RefSeq, Ensembl, and UCSC Genes. We further used samtools to extract uniquely mapped reads.

### Calling RNA variants

RNA variants were called as we described in Additional file 1: Fig. S1. In brief, we detected nucleotide variants between RNA-seq data and the reference genome in each sample. We took variant positions with the mismatch supported by two or more reads

with a base quality score of ≥ 20 and a mapping quality score ≥ 20. Variants were divided into Alu and non-Alu regions. Non-Alu sites were subject to a more stringent variant call as previously described [11]. Last, to facilitate a fair comparison between non-SNP and SNP editing sites in our analyses, we required a minimum number of reference and altered nucleotides ≥ 3 for both non-SNP and SNP editing site call. We inferred the strand information of the sites based on the strand of the genes.

Notably, for SNP editing site call, we considered a site as an authentic SNP editing site only if the corresponding DNA sample had a homozygous genotype. We collected the genotype information from multiple resources and removed a site if its DNA sample had a conflict genotype information between different data resources.

### SNP data

We downloaded all available 1000 Genomes Project and GTEx Project SNP data: (1) Genotype Calls (.vcf) for OMNI SNP Arrays, WES, and WGS of GTEx Project were downloaded from dbGaP database (https://www.ncbi.nlm.nih.gov/gap/); (2) WES data of Geuvadis project were downloaded from EMBL-EBI database (https://www.ebi.ac.uk/arrayexpress/files/E-GEUV-1/genotypes/); (3) Genotype Calls (.vcf) from OMNI SNP Arrays, Affy SNP Arrays, and WGS of the 1000 Genomes Project were downloaded from ftp website (http://ftp.1000genomes.ebi.ac.uk/vol1/ftp/). We discarded all insertion and deletion polymorphisms, SNPs with more than two alleles, SNPs monomorphic (that is, having only one allele) in all populations, and SNPs that did not map uniquely to the human genome (hg19).

### Editing level analysis in different cell lines

RNA-seq data of ADAR1 or ADAR2 overexpressed HEK293 cells and control cells were obtained from Song et al. [27]. RNA-seq data of ADAR1 knockout HEK293 cells and control cells were obtained from Song et al. [28]. HEK293 whole genome sequencing data were obtained from NCBI SRA (SRR2123657). RNA-seq data were trimmed with cutadapt (-q 20,20 --trim-n -m 15) and mapped with HISAT2 [52]. DNA-seq data were trimmed with cutadapt (-q 20,20 --trim-n -m 15) and mapped with BWA (aln -n 6 -t 20). Editing levels of SNP and non-SNP editing sites were called from the mapped data. To compare the editing levels between different samples, we required that sites were covered by at least 10 reads in all samples and the editing level was > 0.02 in at least one sample. In addition, for SNP editing sites, we required that the sites were covered by at least 5 reads in DNA-seq data and no G reads were observed.

RNA-seq data of ADAR1 or ADAR2 knockdown B cells and control cells were obtained from Wang et al. [29]. Editing levels were called as above. For SNP editing sites, we required that the genotype was homozygous (AA) based on the corresponding genome sequence data (GM12004 and GM12750) from the 1000 Genomes Project.

### RNA secondary structure analysis

Two methods were applied to examine the RNA structure. In the first method [25], to identify putative ECS of a given editing site, we searched for the energetically most favorable hybridization region between the editing region (editing site and flanking ± 15 nt) and the extended surrounding region (± 2500 nt around the editing site) using

RNAplex [53]. We required that the extended surrounding region should be within a gene based on known human gene models. As a control, we shuffled the editing region 10,000 times and calculated the mean value of the lowest hybridization energies. For an ECS of a given editing site, if the hybridization energy between the editing region and the ECS was among the top 100 lowest hybridization energies of the shuffled sequences (i.e., $p < 0.01$), we considered it as an ECS with statistical significance.

In the second method [26], to detect the dsRNA structure formed around a given editing site, we aligned the editing region (editing site and flanking ± 25 nt) to the genomic sequence ± 2000 nt of the SNP editing site. We required that the genomic sequence ± 2000 nt of the SNP editing site should be within a gene based on known human gene models. We used bl2seq, with parameters -F F -W 7 -r 2, to align the sequence, and the best alignment score was obtained. As a control, we shuffled the editing region 10,000 times and calculated the mean value of the best scores.

## GO term analysis

GO term enrichment analysis was performed using R package clusterProfiler [54].

## GWAS enrichment analysis

A total of 85 GWAS datasets with full GWAS statistics provided in GWAS catalog (https://www.ebi.ac.uk/gwas/downloads/summary-statistics) were manually checked, and the ones that are not disease-relevant were excluded. Finally, 45 datasets that represent 33 types of diseases were downloaded. For each GWAS dataset, we examined the percentage of editing SNPs that are overlapped with GWAS SNPs with $p$ value < 0.001(%editing_SNP). As a control, we examined the percentage of SNPs with $p$ value < 0.001(%control_SNP). Last, the enrichment score was defined as %editing_SNP/%control_SNP. For a disease with multiple datasets, the dataset with the median enrichment score was shown.

## DAF analysis

For each SNP, we extracted the ancestral allele information and DAF from the VCF files of the 1000 Genomes Project. Bi-allelic SNPs were obtained by VCFtools (--min-alleles 2 --max-alleles 2). Only SNPs with a minor allele frequency (MAF) > 0.001 were used for DAF spectrum analysis.

## Tajima's *D* calculation

Tajima's $D$ was calculated by VariScan [55]. In brief, SNP sites and the flanking 300-bp sequences were used for calculation. Only the SNPs with the flanking sequences located in the same functional classes, such as 3′UTR, 5′UTR, CDS, or ncRNA, were selected. RunMode 12 was chosen to calculate Tajima's $D$.

## Nucleotide diversity calculation

We applied two methods to calculate nucleotide diversity (pi). In Fig. 5b, VariScan was used to calculate nucleotide diversity of a 300-bp region surrounding the editing SNPs. The parameters are the same as the ones used for Tajima's $D$ calculation.

In Fig. 5c, nucleotide diversity of sliding window analysis was performed using VCFtools. The parameters "--window-pi" and "--window-pi-step" were set to 200 bp and 100 bp, respectively.

### Acquisition of T1 and T2 scores

T1 and T2 statistics were obtained from DeGiorgio et al. [42].

### LD analysis

The software PopLDdecay [56] was applied to calculate LD ($r^2$) of pairs of A/G editing SNPs and pairs of control SNPs, using VCF data of the 1000 Genomes Project as the input.

### RNA editing SNP analysis in *D. melanogaster*

The genotype data of 205 *D. melanogaster* inbred lines were downloaded from the Drosophila Genetic Reference Panel Project [43] (http://dgrp2.gnets.ncsu.edu/). The pairwise *D. melanogaster*/*D. simulans* alignment files were from UCSC (http://hgdownload.soe.ucsc.edu/goldenPath/dm3/vsDroSim1/). The list of *D. melanogaster* RNA editing sites was from RADAR database [40]. The ancestral allele of the SNPs was inferred from the homologous *D. simulans* sequence. DAF was calculated based on the genotype information of the 205 inbred lines.

## Supplementary Information

---

**Additional file 1: Fig. S1.** The pipeline of non-SNP and SNP editing site identification. **Fig. S2.** The identification of non-SNP and SNP editing sites. **Fig. S3.** Triplet motif analysis of SNP editing sites. **Fig. S4.** The verification of SNP editing sites. **Fig. S5.** Comparison of the ratios of RNA editing SNP and control SNP. **Fig. S6.** ADAR2 and CASQ2 expression in different types of human tissues. **Fig. S7.** The cumulative distribution of DAF. **Fig. S8.** Sliding window analysis of pi for A/G editing SNPs. **Fig. S9.** non-Alu editing SNPs are subject to a stronger balancing selection compared with Alu editing SNPs. **Fig. S10.** DAF analysis reveals RNA editing SNPs as the target of balancing selection in flies.

**Additional file 2: Table S1.** The list of GTEx data used for analysis.

**Additional file 3: Table S2.** SNP and non-SNP RNA editing sites.

**Additional file 4: Table S3.** *P* values of DAF analysis.

**Additional file 5: Table S4.** T1,T2 statistics of RNA editing SNPs.

**Additional file 6.** Review history.

---

### Acknowledgements

We thank Jin Billy Li and Tao Sun for the discussion of the manuscript and Michael DeGiorgio for sharing the T1 and T2 score data. The Genotype-Tissue Expression (GTEx) project was supported by the Common Fund of the Office of the Director of the National Institutes of Health (commonfund.nih.gov/GTEx). Additional funds were provided by the National Cancer Institute (NCI); National Human Genome Research Institute (NHGRI); National Heart, Lung, and Blood Institute (NHLBI); National Institute on Drug Abuse (NIDA); National Institute of Mental Health (NIMH); and National Institute of Neurological Disorders and Stroke (NINDS). Donors were enrolled at Biospecimen Source Sites funded by NCISAIC-Frederick, Inc. (SAIC-F) subcontracts to the National Disease Research Interchange (10XS170) and Roswell Park Cancer Institute (10XS171). The Laboratory, Data Analysis, and Coordinating Center (LDACC) was funded through a contract (HHSN268201000029C) to The Broad Institute, Inc. Biorepository operations were funded through an SAIC-F subcontract to Van Andel Institute (10ST1035). Additional data repository and project management were provided by SAIC-F (HHSN261200800001E). The Brain Bank was supported by a supplement to University of Miami grant DA006227.

### Peer review information

### Review history

The review history is available as Additional file 6.

## Authors' contributions
H.Z. and R.Z. conceived the project. H.Z., Q.F., and X.R.S. contributed to the computational analyses. Q.F., Z.Q.P., and Z.C.H. contributed to the data collection. H.Z., X.R.S., and R.Z. wrote the paper with input from T.T. and X.L.H.. The authors read and approved the final manuscript.

## Funding
This study was supported by grants from National Natural Science Foundation of China (91631108 and 31571341 to R.Z.), Guangdong Innovative and Entrepreneurial Research Team Program (2016ZT06S638 to R.Z.), and the State Key Laboratory of Genetic Resources and Evolution (GREKF18-08).

## Availability of data and materials
Geuvadis RNA-seq data can be obtained from the Geuvadis consortium (https://www.internationalgenome.org/data-portal/data-collection/geuvadis). Geuvadis genotype data can be obtained from the 1000 Genomes Project (http://www.internationalgenome.org/data/). GTEx RNA-seq and Genotype data can be obtained from the GTEx consortium (https://gtexportal.org/home/). GWAS datasets with full GWAS statistics can be obtained from GWAS catalog (https://www.ebi.ac.uk/gwas/downloads/summary-statistics). Fly RNA editing site annotations are available at the RADAR website (http://rnaedit.com/download/).

## Ethics approval and consent to participate
Not applicable

## Competing interests
The authors declare no competing financial interests.

## Author details
[1]Key Laboratory of Gene Engineering of the Ministry of Education, State Key Laboratory of Biocontrol, School of Life Sciences, Sun Yat-Sen University, Guangzhou, People's Republic of China. [2]State Key Laboratory of Genetic Resources and Evolution, Kunming Institute of Zoology, Chinese Academy of Sciences, Kunming, People's Republic of China. [3]State Key Laboratory of Biocontrol, School of Life Sciences, Sun Yat-Sen University, Guangzhou, People's Republic of China. [4]RNA Biomedical Institute, Sun Yat-Sen Memorial Hospital, Sun Yat-Sen University, Guangzhou, People's Republic of China.

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

## 