## [**Additional file 6.** Review history. · Genome Biology]

Review History

First round of review

Reviewer 1

Are you able to assess all statistics in the manuscript, including the appropriateness of statistical tests used? Yes, and I have assessed the statistics in my report.

Comments to author:

The manuscript "Human A-to-I RNA editing SNP loci are enriched in GWAS signals for autoimmune diseases and under balancing selection" by Zhang et al deals with the interesting cases of A-to-I RNA editing sites that overlap, in the same genomic location, with SNPs. The most interesting finding in this work is the discovery that some of the A/G editing SNPs are highly enriched in GWAS signals of autoimmune and immune-related diseases.

While the work is very interesting, timely and represent a significant advance over previously published studies, I believe few adjustments and modifications are needed before it can be acceptable for publication. Addressing the following points is required in order to strength the validity of some of the key conclusions:

The main challenge this and related RNA editing project is how to distinguish between real editing events and those who are the results of technical artifacts, many of which are known to be the source of false identifications of editing events (mutations, polymorphisms sequencing and alignment errors). The ability to evaluate the success of the editing predication rate can get complicated as there are two main classes of A-to-I editing events in the human genome: almost all sites are located within Alu repeats, and it is now safe to say that virtually any "A" in Alu can be edited. As expected, Alu sites, have features such as ADAR sequence motif that support their validity. Thus the validity of each of such site is very high. On the other hand, sites that are located in non-Alu genomic regions are much more difficult to detect correctly. As this sites includes the more interesting and important set of sites of these paper, it will be very important to show that this set of sites (and especially the CDS ones) have the ADAR sequence motif. Moreover, a critical control is to check if A-G editing SNP are more common than C/T editing SNP (while applying the same pipeline as for A-G) when focusing in only CDS regions that are not overlap Alu.

Without this two critical tests, the level of confidence for the sets of editing -SNPs that are located not within Alu is rather low.

Additional comments:

-How many SNP-editing are located within Alu? (some of the CDS sites are probably overlap Alu exons)
-Some support for the low reliability of some of the key sites appear in the manuscript are coming from close inspection of their genomic sequence. For example, the flanking sequence of rs2241880 has 100% sequence identity to a known polymorphic genomic locus (chr2_KN538363v1_fix), thus probably not a real RNA editing event.

-Can the authors give evidence for the secondary structure needed for ADAR activity for the non Alu sites?

-"These analyses confirmed that both non-SNP and SNP editing sites were 105 real editing events."- No, its only support the idea that there is an enrichment (see comments above).

-An early paper that deal with same class of editing sites, entitled "Identification of RNA editing sites in the SNP database" (PMID 16100382) should be discuss.

-Line 89:" To identify editing sites with high confidence, we required that the fractions of A-to-G/T-to-C match are >80%" not clear.

Reviewer 2

Are you able to assess all statistics in the manuscript, including the appropriateness of statistical tests used? No, I do not feel adequately qualified to assess the statistics.

Comments to author:

Zhang et al. study a heretofore unexplored subset of human ADAR A-to-I editing sites that overlap with known SNPs. They developed a novel RNA-seq analysis pipeline for the identification of these SNPs and demonstrate their authenticity across two data sets (GTEx and Geuvadis) by demonstrating close adherence to typical A-to-I editing site indicators. They go on to characterize the genic location of all identified sites, finding that SNP edit sites are enriched in the CDS compared to non-SNP edit sites, suggesting that SNP editing sites may be functionally significant. They also find that the editing SNPs were biased toward A/G (or T/C) genotypes compared to control SNPs, an indication that the G allele (equivalent to edited T) was selected to be maintained. This is a surprising finding, and the authors further investigate the functional significance.

Zhang et al. also probe the importance of these SNPs in disease, reporting that they are enriched in GWAS loci associated with autoimmune and immune-related pathogenesis. The authors demonstrate that in one of these, inflammatory bowel disease (IBD), the two leading SNPs of GWAS fine-mapping data are editing SNPs. This supports their conclusion that editing SNPs could play a salient role in the development of disease. Finally, Zhang et al. use statistical methods from population genetics to determine if these SNPs are preserved in the population via balancing selection. Derived allele frequency analysis, Tajima's D, nucleotide diversity, T1/T2 scores, and LD were employed, and all results suggest that balancing selection is indeed occurring. Ultimately, the findings that balancing selection is responsible for maintaining these SNPs in the population and that these SNPs are potentially related to immune function makes a case that editing SNPs play a part in the maintenance of human immunogenetic plasticity. While the paper uncovers a striking and novel finding, it lacks sufficient discussion regarding the implications of all its findings and there are multiple places where additional analysis could be conducted to confirm what the statistics suggest. As such, we recommend major revision.

Major comments:

- 1) To broaden the scope of the paper, it is recommended that the authors investigate whether editing SNPs are under balancing selection in other species, such as mice. Were a mouse SNP database given the same statistical treatment as the data in Figure 4 and it was found that mouse editing SNPs were also maintained in the population at intermediate levels, the authors could point to editing SNPs as a generally conserved genetic feature.
- 2) It is suggested that the authors comment on the editing frequency between SNP edit sites, non-SNP edit sites, and GWAS SNP edit site. This would further allow the readers to assess how biologically significant the SNP edits are. For example, is the edit frequency at SNP edits generally low, or high (more than 50%)? How does editing frequency at SNP edit sites compare to non-SNP edit sites?
- 3) The authors demonstrate that editing SNPs are enriched with GWAS loci associated with autoimmune/immune-related disease. If the scope of Figure 2a were expanded to include many more (or all) diseases in the GWAS database, would editing SNPs be generally enriched with GWAS SNPs associated with human disease? Or, would there be a similar distribution of diseases with an enrichment score above 0 and below 0?
- 4) To validate the findings and interpretation in Figure 2b, can the authors express the un-edited transcript of ATG16L1 with ADAR1, and confirm that the SNP edits, do truly get edited? Furthermore, can they show that the edited transcript has a phenotype as shown with the ATG16L1 T300A loci?

Minor comments:

- 1) Please comment on why SNP edits increase in the anti-sense transcript in Figure 1e. This is very

interesting.

- 2) In Figure 1e there is an enrichment of SNP editing sites over non-SNP editing sites in the CDS. What do the authors make of this, and could this proportional increase in SNP editing sites be due to an overall increased proportion of SNPs in CDS regions over non-CDS regions?
- 3) What are the total number of SNP edits vs. non-SNP edit sites the authors uncovered in the GTEX and Geuvadis database? It would be great to have an idea of the proportion of SNP edits, among all the edits.
- 4) The inclusion of a supplementary figure which shows the cumulative density functions used to calculate the Kolmogorov-Smirnov p-values in Figure 3 would be useful, as the current presentation of data does not provide a strong visualization of the statistical significance in each plot. The modified y-axis scale in Figure 3d obscures this point further.
- 5) The authors claim that edits in the 3'UTR (a highly ADAR1 edited genic region) are disproportionately present in genes that mediate innate immune stress (Figure 1H). They claim this coincides with ADAR1's role as a stress response suppressor. This assertion is inconsistent with accepted mechanisms for how ADAR1 functions, as its anti-inflammatory properties are not understood to arise from editing the transcripts of stress-associated genes. In other words, ADAR1 does not regulate inflammation (stress) by editing stress related genes. ADAR1 is thought to regulate inflammation by editing dsRNA structures (that rise from various types of genes, not necessarily stress related genes) that can potentially activate innate immune sensors.

We are very grateful to all the reviewers for the constructive feedback and the opportunity to improve our work. We have revised our manuscript accordingly and highlighted the major changes in yellow for easy tracking. Please see below for our point-by-point response to reviewers' comments.

Reviewer reports:

Reviewer #1: The manuscript "Human A-to-I RNA editing SNP loci are enriched in GWAS signals for autoimmune diseases and under balancing selection" by Zhang et al deals with the interesting cases of A-to-I RNA editing sites that overlap, in the same genomic location, with SNPs. The most interesting finding in this work is the discovery that some of the A/G editing SNPs are highly enriched in GWAS signals of autoimmune and immune-related diseases.

While the work is very interesting, timely and represent a significant advance over previously published studies, I believe few adjustments and modifications are needed before it can be acceptable for publication. Addressing the following points is required in order to strength the validity of some of the key conclusions:

The main challenge this and related RNA editing project is how to distinguish between real editing events and those who are the results of technical artifacts, many of which are known to be the source of false identifications of editing events (mutations, polymorphisms sequencing and alignment errors). The ability to evaluate the success of the editing predication rate can get complicated as there are two main classes of A-to-I editing events in the human genome: almost all sites are located within Alu repeats, and it is now safe to say that virtually any "A" in Alu can be edited. As expected, Alu sites, have features such as ADAR sequence motif that support their validity. Thus the validity of each of such site is very high. On the other hand, sites that are located in non-Alu genomic regions are much more difficult to detect correctly. As this sites includes the more interesting and important set of sites of these paper, it will be very important to show that this set of sites (and especially the CDS ones) have the ADAR sequence motif. Moreover, a critical control is to check if A-G editing SNP are more common than C/T editing SNP (while applying the same pipeline as for A-G) when focusing in only CDS regions that are not overlap Alu. Without this two critical tests, the level of confidence for the sets of editing -SNPs that are located not within Alu is rather low.

We thank the reviewer for this comment. To address the reviewer's question, we have now examined the Non-Alu CDS sites (518 sites) and the remaining sites (7272 sites) separately. We found that non-Alu CDS sites had a slightly weaker ADAR motif than other sites (**Figure S3**). This result suggests that non-Alu CDS sites may have a higher false-discovery rate than sites in other genic regions and we have clarified this in the revision (**Page 5, Lines 121-123**).

Moreover, we applied the same pipeline to call C-to-T/G-to-A editing SNPs in the

GTEX and Geuvadis datasets. We identified 11 and 0 sites (including 3 and 0 non-Alu sites in the CDS regions), which were much less than the 6,407 and 1,651 A-to-G sites we identified (including 460 and 68 non-Alu sites in the CDS regions).

Figure S3. Triplet motif analysis of SNP editing sites. Because the Geuvadis dataset had a limited number of non-Alu CDS sites, we merged the SNP editing site lists from the GTEX and Geuvadis datasets for analysis. Non-Alu CDS sites (518 sites) and the remaining sites (7272 sites) were analyzed separately.

Additional comments:

-How many SNP-editing are located within Alu? (some of the CDS sites are probably overlap Alu exons)

There are 2,841 SNP editing sites within Alu and 9 of them are in the CDS regions. We have added the number in **Fig. 2b**.

Fig. 2b. The proportion of SNP and non-SNP editing sites in Alu and non-Alu regions of different genic locations. Numbers of editing sites are listed above the bars.

-Some support for the low reliability of some of the key sites appear in the manuscript are coming from close inspection of their genomic sequence. For example, the flanking sequence of rs2241880 has 100% sequence identity to a known polymorphic genomic locus (chr2_KN538363v1_fix) , thus probably not a real RNA editing event.

We thank the reviewer for pointing this out. The result related to this SNP has been removed in the revision.

-Can the authors give evidence for the secondary structure needed for ADAR activity for the non Alu sites?

We thank the reviewer for this suggestion. To address the reviewer's question, we performed two analyses.

First, we predicted the editing complementary sequence (ECS) of each editing region (SNP editing site and flanking ± 15 nt) and the shuffled editing region, as previously described[Licht K et al., Genome Res 2019]. Next, we compared the hybridization energies between SNP editing regions and their predicted ECSs with those between shuffled editing regions and their predicted ECSs. We found significantly lower hybridization energies of editing regions than the shuffled regions (**Fig. 1d**). Moreover, about 44% of the SNP editing sites had a statistically significant ECS.

Second, we detected the potential dsRNA structures containing SNP editing sites using bl2seq, as previously described[Porath HT et al., Genome Biology 2017]. We found that the editing regions formed dsRNA structures with significantly higher alignment scores as compared to the shuffled regions (**Fig. 1e**). The same analyses were performed for non-Alu SNP editing sites and the conclusions still held (**Figure S4a-b**).

Figure 1

Figure S4

Fig.1d. Comparison of the hybridization energies between SNP editing regions (SNP editing sites and flanking ± 15 nt) and their predicted complementary sequences with those between shuffled editing regions and their predicted complementary sequences (**Methods**). For each SNP editing site, we shuffled the editing region and predicted its complementary sequence. We repeated this 10,000 time and the mean value was calculated. The p value was calculated with the Pairwise Wilcoxon Rank Sum Test.

Fig.1e. Comparison of the BLAST Scores between SNP editing regions (SNP editing sites and flanking ± 25 nt) and shuffled editing regions. BLAST Score represents the overall quality of an alignment (aligning the editing region to the genomic sequence ± 2000 nt of the SNP editing site, **Methods**). The p value was calculated with the Pairwise Wilcoxon Rank Sum Test.

Figure S4a. Comparison of the hybridization energies between non-Alu SNP editing regions and their predicted complementary sequences with those between shuffled editing regions and their predicted complementary sequences. The analysis was performed as in **Fig. 1d**. About 15.2% of the SNP editing sites had a statistically significant ECS (**Methods**).

Figure S4b. Comparison of the BLAST Scores between non-Alu SNP editing regions and shuffled editing regions. The analysis was performed as in **Fig. 1e**.

- "These analyses confirmed that both non-SNP and SNP editing sites were real editing events." - No, its only support the idea that there is an enrichment (see comments above).

Thanks for pointing it out. We have rephrased the text accordingly (**Page 5, lines 119-121**):

“These analyses support that both non-SNP and SNP RNA variants we called were enriched in authentic editing events.”

-An early paper that deal with same class of editing sites, entitled "Identification of RNA editing sites in the SNP database" (PMID 16100382) should be discuss.

We thank the reviewer for this suggestion. We have carefully read the suggested article and added this early observation in the introduction section (**Page 3-4, Lines 74-79**):

“Before the next-generation sequencing era, a pioneer study has identified A-to-I RNA editing sites in the SNP database[21]. In many cases, SNPs overlapped with editing sites are annotated using expressed sequence tags, and thus are RNA editing sites instead of SNPs. However, it is possible that some of these SNPs are real SNPs that can be edited. And such editing SNPs are of importance for functional and evolutionary studies of RNA editing.”

-Line 89:" To identify editing sites with high confidence, we required that the fractions of A-to-G/T-to-C match are >80%" not clear.

We apologize for not clearly describing how the cutoff was set. We have rephrased the sentence accordingly:

“To identify editing sites with high confidence, we only selected samples in which the proportion of A-to-G/T-to-C variants to total variants were at least 80% for editing site call.”

Reviewer #2: Zhang et al. study a heretofore unexplored subset of human ADAR A-to-I editing sites that overlap with known SNPs. They developed a novel RNA-seq analysis pipeline for the identification of these SNPs and demonstrate their authenticity across two data sets (GTEx and Geuvadis) by demonstrating close adherence to typical A-to-I editing site indicators. They go on to characterize the genic location of all identified sites, finding that SNP edit sites are enriched in the CDS compared to non-SNP edit sites, suggesting that SNP editing sites may be functionally significant. They also find that the editing SNPs were biased toward A/G (or T/C) genotypes compared to control SNPs, an indication that the G allele (equivalent to edited 'I') was selected to be maintained. This is a surprising finding, and the authors further investigate the functional significance.

Zhang et al. also probe the importance of these SNPs in disease, reporting that they are enriched in GWAS loci associated with autoimmune and immune-related

pathogenesis. The authors demonstrate that in one of these, inflammatory bowel disease (IBD), the two leading SNPs of GWAS fine-mapping data are editing SNPs. This supports their conclusion that editing SNPs could play a salient role in the development of disease. Finally, Zhang et al. use statistical methods from population genetics to determine if these SNPs are preserved in the population via balancing selection. Derived allele frequency analysis, Tajima's D, nucleotide diversity, T1/T2 scores, and LD were employed, and all results suggest that balancing selection is indeed occurring. Ultimately, the findings that balancing selection is responsible for maintaining these SNPs in the population and that these SNPs are potentially related to immune function makes a case that editing SNPs play a part in the maintenance of human immuno-genetic plasticity. While the paper uncovers a striking and novel finding, it lacks sufficient discussion regarding the implications of all its findings and there are multiple places where additional analysis could be conducted to confirm what the statistics suggest. As such, we recommend major revision.

Major comments:

1) To broaden the scope of the paper, it is recommended that the authors investigate whether editing SNPs are under balancing selection in other species, such as mice. Were a mouse SNP database given the same statistical treatment as the data in Figure 4 and it was found that mouse editing SNPs were also maintained in the population at intermediate levels, the authors could point to editing SNPs as a generally conserved genetic feature.

We thank the reviewer for this great suggestion. Because only a limited number of mouse strains have genotype data, we examined the evolution of editing SNPs in *D. melanogaster*. The genotype data from the Drosophila Genetics Reference Panel Project (DGRP), which consists 205 sequenced inbred lines derived from Raleigh (NC), United States, were used to perform SNP allele type and DAF analysis. We examined all known RNA editing sites from RADAR2 database. A total of 743 sites were found to be overlapped with SNPs in DGRP. Similar to human, fly editing SNPs were biased toward A/G or T/C genotypes as compared with the control SNPs (**Figure S10a**). Moreover, the DAF distribution of A/G editing SNPs was significantly skewed toward intermediate frequency alleles in all functional classes relative to intergenic regions (**Figure S10b-c**, p values in **Table S3**). In contrast, a shift in a DAF distribution toward low frequency alleles was observed for non-A/G editing SNPs, which is indicative of negative selection (**Figure S10d-e**, p values in **Table S3**). These results support that RNA editing as the target of balancing selection in flies.

Figure S10a. Comparison of the SNP types between editing SNPs and control SNPs. All SNPs in the DGRP dataset that are with A or T as the reference allele were selected as control SNPs.

Figure S10b-c. DAF distributions and cumulative distributions of DAF for A/G editing SNPs and intergenic control SNPs. P values were calculated with the Kolmogorov-Smirnov test by comparing the DAF distribution of editing SNPs in a defined genic location with the distribution of SNPs in intergenic regions (Table S3). The numbers of RNA editing SNPs in each genic location are shown in parentheses. Intergenic-A/G: all A/G SNPs located in the intergenic regions.

Figure S10d-e. DAF distributions and cumulative distributions of DAF for non-A/G editing SNPs and intergenic control SNPs. Because of the limited number of non-A/G editing SNPs, all SNPs were combined for analysis. Intergenic-non-A/G: all non-A/G SNPs located in the intergenic regions.

2) It is suggested that the authors comment on the editing frequency between SNP edit sites, non-SNP edit sites, and GWAS SNP edit site. This would further allow the readers to assess how biologically significant the SNP edits are. For example, is the edit frequency at SNP edits generally low, or high (more than 50%)? How does editing frequency at SNP edit sites compare to non-SNP edit sites?

To address the reviewer's question, we compared the editing levels between these three classes of sites (Fig. 3b). We found that SNP and GWAS SNP editing sites in CDS regions had higher editing levels than non-SNP editing sites, while SNP editing

sites in intronic or intergenic regions had similar editing levels as compared with non-SNP editing sites (**Fig. 3b**). Thus it seems that SNP editing sites in functionally important regions tended to have higher editing levels.

Figure 3b. Comparison of editing levels between SNP editing sites, non-SNP editing sites, and GWAS SNP editing sites. 5'UTR sites are not shown because only 3 GWAS SNP sites were found. For this analysis, we used the representative editing level of each editing site, which is the maximum editing level across all GTEx tissue types we profiled. The editing level of a tissue is the mean editing level of all samples in a given tissue. P values were calculated with the Mann-Whitney U test.

3) The authors demonstrate that editing SNPs are enriched with GWAS loci associated with autoimmune/immune-related disease. If the scope of Figure 2a were expanded to include many more (or all) diseases in the GWAS database, would editing SNPs be generally enriched with GWAS SNPs associated with human disease? Or, would there be a similar distribution of diseases with an enrichment score above 0 and below 0?

We apologize for not clearly describing how we collected the GWAS datasets. We have actually included all types of human disease GWAS data we can collect in our enrichment analysis.

In brief, a total of 85 GWAS datasets with full GWAS statistics provided in GWAS catalog (<https://www.ebi.ac.uk/gwas/downloads/summary-statistics>) were manually checked and the ones that are not disease-relevant were excluded. Finally, 45 datasets that represent 33 types of diseases were downloaded. For a disease with multiple datasets, the dataset with the median enrichment score was shown. We have clarified this in the revision (**Page 24-25, Lines 552-555**).

4) To validate the findings and interpretation in Figure 2b, can the authors express the un-edited transcript of ATG16L1 with ADAR1, and confirm that the SNP edits, do truly get edited? Furthermore, can they show that the edited transcript has a phenotype as shown with the ATG16L1 T300A loci?

As the reviewer 1 pointed out that ATG16L1 is located in a known polymorphic genomic locus and probably not a real RNA editing event, we have decided to remove the result related to this SNP in the revision.

To provide additional experimental evidence to demonstrate that most SNP editing sites we identified are authentic editing events, we examined their editing level changes upon overexpression or knockout/knockdown of ADARs. We found that both non-SNP and SNP editing sites had increased levels upon overexpression of ADAR1 or ADAR2 in HEK293 cells (**Fig. 1f**). We also examined ADAR1 knockout HEK293 cells and ADAR1 or ADAR2 knockdown B cells. Both non-SNP and SNP editing sites had decreased editing levels in the knockout or knockdown cells (**Fig.1g** and **Figure S4c-d**).

Figure 1

Figure S4

Fig. 1f. Boxplots showing the editing level changes of SNP editing sites and non-SNP editing sites upon ADAR1 or ADAR2 overexpression in HEK293 cells.

Fig. 1g. Boxplots showing the editing level changes of SNP editing sites and non-SNP editing sites between wild-type and ADAR1 knockout HEK293 cells.

Figure S4c. Boxplots showing the editing level changes of SNP editing sites and non-SNP editing sites upon ADAR1 or ADAR2 knockdown in B cells (GM12004). Data were from Wang et al.

Figure S4d. Boxplots showing the editing level changes of SNP editing sites and non-SNP editing sites upon ADAR1 knockdown at different time points in B cells (GM12750). ADAR1 KD-1, ADAR1 KD-2, ADAR1 KD-3, and ADAR1 KD-4: 24h, 48h, 72h, and 96h after the siRNA transfection. Data were from Wang et al.

Minor comments:

1) Please comment on why SNP edits increase in the anti-sense transcript in Figure 1e. This is very interesting.

We thank the reviewer for this suggestion. We agree with the reviewer that this

observation is very interesting. On the one hand, antisense transcripts may form dsRNA structures by themselves and be edited by ADARs. On the other hand, once co-expressed, sense-antisense pairs may form perfectly-matched dsRNA structures, which are likely ADAR substrates. Our preliminary analysis suggests that such antisense editing events might be a group of previously overlooked editing events with functional relevance, which are worthwhile for future study. Because we are aware that the disease-relevance and functional significance of sense-antisense pairs have been characterized in a much more comprehensive and detailed way in a manuscript from Dr. Jin Billy Li's lab and their manuscript is currently under consideration (personal communication), we have decided not to further discuss this issue in the text.

2) In Figure 1e there is an enrichment of SNP editing sites over non-SNP editing sites in the CDS. What do the authors make of this, and could this proportional increase in SNP editing sites be due to an overall increased proportion of SNPs in CDS regions over non-CDS regions?

The densities of editing SNPs varied among different functional classes, and such difference is not due to the difference of background SNP densities in different functional classes (**Figure S5**). We have clarified this in the revision (**Page 6, Lines 157-159**).

Figure S5. Comparison of the ratios of RNA editing SNP and control SNP. For RNA editing SNPs, the ratio was defined as the number of editing SNPs divided by the number of non-SNP editing sites. As a control, we calculated the ratio of control SNPs, which was defined as the number of SNPs with “A” as the reference allele divided by the total number of “A” in the given functional class. Notably, because the SNPs themselves had no strand information, we were unable to calculate the ratios of control SNPs in antisense transcripts and intergenic regions.

3) What are the total number of SNP edits vs. non-SNP edit sites the authors uncovered in the GTEx and Geuvadis database? It would be great to have an idea of the proportion of SNP edits, among all the edits.

For the GTEx dataset, we identified 6,407 SNP editing sites and 259,462 non-SNP editing sites. For the Geuvadis dataset, we identified 1,651 SNP editing sites and 34,419 non-SNP editing sites. The proportions of SNP editing sites in the GTEx and Geuvadis datasets are 0.024 and 0.046, respectively. We have added this information in the revision (**Page 4, Lines 98-102**).

4) The inclusion of a supplementary figure which shows the cumulative density functions used to calculate the Kolmogorov-Smirnov p-values in Figure 3 would be useful, as the current presentation of data does not provide a strong visualization of the statistical significance in each plot. The modified y-axis scale in Figure 3d obscures this point further.

Thanks. We have included the CDF plots in the revision (**Figure S7**).

Figure S7a-b. The cumulative distributions of DAF for A/G (a) or non-A/G (b) editing SNPs and intergenic control SNPs used in Fig. 4a-b.

Figure S7c-d. The cumulative distributions of DAF for A/G editing SNPs with A (c) or G (d) allele as the ancestral allele and intergenic control SNPs used in Fig. 4c-d.

Figure S7e. The cumulative distributions of DAF for non-A/G editing SNPs and intergenic control SNPs used in Fig. 4e.

Figure S7f. The cumulative distributions of DAF for the SNPs located in the upstream and downstream of 3' UTR non-SNP editing sites and intergenic control SNPs used in Fig. 4f.

5) The authors claim that edits in the 3'UTR (a highly ADAR1 edited genic region) are disproportionately present in genes that mediate innate immune stress (Figure 1H). They claim this coincides with ADAR1's role as a stress response suppressor. This assertion is inconsistent with accepted mechanisms for how ADAR1 functions, as its anti-inflammatory properties are not understood to arise from editing the transcripts of stress-associated genes. In other words, ADAR1 does not regulate inflammation (stress) by editing stress related genes. ADAR1 is thought to regulate inflammation by editing dsRNA structures (that rise from various types of genes, not necessarily stress related genes) that can potentially activate innate immune sensors.

We thank the reviewer for pointing this out. This claim has been removed in the revision.

Second round of review

Reviewer 1

The authors address nicely almost all the concerns raised by myself and the other referee. Yet, as evident from the case of rs2241880, it will be important to clean the dataset further by systematic removal of similar cases (SNPs that are evidently have 100% sequence identity of the flanking sequence to a known polymorphic genomic locus) or in general any possible case of mapping issues- where the validity of the SNP and/or the editing sites is at question- mainly relevant where the genomic region of a SNP has a very close sequence identity to another location in the genome/transcriptome.

Authors' response

We also addressed the reviewer's concern by adding a column to Table S2 to mark the sites (54 sites in total) that are similar to the cases of rs2241880.